# Rare event detection by progressive clustering undersampling

**Amr Abuzeid**●*, **Elena Jolkver**●

Data Science Dept/IU Internationale Hochschule GmbH, Juri-Gagarin-Ring, Erfurt, Germany

* amr.abuzeid@iu-study.org

## Abstract

Capturing rare events in severely imbalanced datasets is challenging, as the learning and optimization processes are often biased toward the majority class. To address this issue, this study explores various resampling techniques and introduces a novel method called Progressive Clustering Undersampling (PCU). This technique removes negative instances that are distant from positive ones. PCU was compared with eight common undersampling and two oversampling techniques, consistently outperforming them on highly imbalanced and noisy datasets.

The workflow demonstrates that rare anomalies can be effectively predicted using unsupervised methods based on frequency-driven decision boundaries. Progressive clustering ultimately identifies clusters with the highest concentration of positive instances. These delineated clusters are then saved by supervised models and used in the preparatory phase before prediction. The proposed method produces two outputs: one optimized for a high F1-score and the other for high precision. Overall, this approach presents a promising solution for identifying rare anomalies in complex, imbalanced data environments.

## 1 Introduction

Detecting minority classes or rare anomalies is essential in many industries due to its critical role in optimizing processes, facilitating critical decision making, and improving the efficiency of resource allocation. In this work, we present a detailed demonstration of a proposed Machine Learning (ML) workflow applied to petroleum exploration as a real-world example case.

Detecting hydrocarbon zones after drilling involves a trade-off between cost, speed, and accuracy. Fast, cost-effective methods offer preliminary insights but may lack precision, while ground truth methods such as core sampling and perforation provide highly accurate results but are expensive and time consuming.

Core sampling involves extracting cylindrical sections of subsurface formations for detailed analysis; while providing invaluable insights into fluid content and flow capacity, it is expensive due to high operational costs for drilling and retrieval, as well

**Data availability statement:** Yes - all data are fully available without restriction.

**Funding:** The author(s) received no specific funding for this work.

**Competing interests:** The authors have declared that no competing interests exist.

as laboratory expenses for fluid saturation tests. Geologists often rely on electrical logging tools, which are lowered into boreholes, to detect hydrocarbons.

Raw data from logging tool measurements is processed through a series of advanced steps that apply mathematical and physical functions, guided by thresholds derived from linear relationships or other geological references. This deterministic method for identifying hydrocarbon reservoirs is about 85-95% reliable, so errors may occur [1].

Perforation of hydrocarbon zones enables hydrocarbons to flow into the wellbore and be produced at the surface. However, engineers must ensure that this costly operation is not used merely to confirm the presence of hydrocarbons. Instead, perforation should be reserved for zones where there is high confidence in hydrocarbon presence, in order to avoid false positives—cases where non-productive zones are mistakenly identified as productive. On the other hand, failing to identify a productive zone (a false negative) can be even more costly, as the well has already been drilled.

Building upon the principles of statistical inference, ML has evolved to uncover complex relationships that deterministic methods might miss. For instance, reservoir properties prediction depends on complex interactions of multiple measurement variables that ML models can uncover. Despite these advantages, rare anomalies, unlike other classification tasks, pose a distinct challenge to handle. The growing focus of research papers on imbalanced classification challenges highlights the critical importance of this issue across a wide range of domains, from management science to engineering. Rare events, unusual patterns, and abnormal behaviors are inherently difficult to detect [2].

Detecting the minority class in imbalanced data is particularly difficult because the learning process and optimization tend to focus more on the majority class. For instance, in a scenario where 90% of the instances belong to the majority class, a model predicting all instances as part of the majority class would achieve 90% accuracy. Imbalanced data can degrade the performance of most standard classification algorithms [3]. In artificial neural networks, optimization during training often favors the majority class, as iterations primarily result in correct classifications. Consequently, in the backpropagation process, the weights are rarely updated, leading to a biased model that disproportionately favors the majority class [4].

Many classifiers depend on assumptions about class distribution and shape, typically expecting balanced class sizes. When a rare or minority class is underrepresented, there may not be enough examples for the model to learn its patterns effectively, leading to poor recognition of such events [5]. For instance, K Nearest Neighbor (KNN) often misclassifies minority class instances because their nearest neighbors usually belong to the majority class. This misclassification becomes worse as the class imbalance increases [6]. Decision trees also struggle with overlapping classes, requiring many tests to correctly separate minority class cases. Although pruning helps eliminate overly specific branches, it tends to favor the majority class by assigning new leaf nodes to it—further marginalizing the minority class in the predictions [6].

Support Vector Classifiers (SVCs) minimize overall error, but with imbalanced data they bias the decision boundary toward the majority class, often causing many false

negatives. In high-stakes applications—such as medical diagnosis or adversarial attack detection—these missed minority-class cases can lead to serious consequences [7]. Thus, effectively addressing this imbalance is crucial for building and evaluating models that can accurately identify minority class instances. The aim of this study is to develop a novel under-sampling technique for prediction of the minority class, particularly in severely imbalanced scenarios or where the anomalies exhibit some level of uncertainty.

The rest of the paper is structured as follows: the following section describes the existing resampling methods to deal with imbalance in ML. Sect 3 outlines the research methodology including relevant algorithms, while Sect 4 and 5 present findings and a synthesis of the implementation of the method across different datasets, discussing and interpreting the effect of noise and balance on different resampling methods. Finally, the conclusion (Sect 6) summarizes the key findings, emphasizing their significance and potential applications.

## 2 Related work

Solutions for addressing class imbalance can be categorized into two types: algorithmic-level methods and data-level methods [8]. In this section, we review representative techniques from both categories, highlighting their strengths and limitations.

### 2.1 Algorithmic methods

Algorithmic-level methods focus on designing or improving classification algorithms to mitigate biases introduced by imbalanced data. These methods encompass techniques such as threshold adjustments, cost-sensitive learning, one-class learning and ensemble learning [8,9].

Bootstrap aggregating (bagging) and boosting are ensemble methods used to address class imbalance [10,11]. Random Forest, a common bagging technique, mitigates imbalance by using bootstrapped subsets that often include minority class samples [12,13]. Despite potential bias in individual trees, the ensemble's voting mechanism balances errors. Boosting methods like Adaptive Boosting (Adaboost) rely on weighted voting based on the accuracy of the learner. Gradient boosting builds on this by incorporating a loss function for weighting predictions. A key implementation, Extreme Gradient Boosting (XGBoost), enhances performance through regularization and efficient computation. According to Elor et al. [14], strong classifiers like XGBoost may reduce the need for explicit balancing techniques. While algorithmic-level methods can partially address class imbalance, such as by assigning higher weights to the minority class, they may not always be sufficient to handle class imbalance without data preprocessing approaches.

### 2.2 Resampling methods

Data-level preprocessing addresses the imbalance either by reducing the number of majority class samples or by increasing the number of minority class samples, known as undersampling and oversampling, respectively.

**2.2.1 Oversampling.** Chawla et al. [15] introduced an oversampling technique called Synthetic Minority Oversampling Technique (SMOTE), which generates synthetic examples instead of merely duplicating existing ones. For each minority class instance, SMOTE selects one or more of its k nearest neighbors and creates synthetic samples by interpolating between the original instance and its neighbors. This process typically leads to improved performance compared to random or simple oversampling methods [6]. SMOTE has been applied in various tasks, including sentence boundary detection in speech and image classification. However, SMOTE has limitations. Synthetic samples are sometimes generated too close to the majority class or near noisy instances, which can negatively affect classification performance. Additionally, SMOTE may perform poorly with high-dimensional data, where distance metrics lose their discriminative power and become less informative.

Adaptive Synthetic Oversampling Approach (ADASYN) is an advanced oversampling method that adaptively generates synthetic data points for the minority class based on the classification difficulty [16]. Unlike methods such as SMOTE,

which generates synthetic samples uniformly, ADASYN focuses on regions where the minority class is harder to classify, tailoring the oversampling process to the local distribution of the data [17].

However, synthetic instances, including those generated by ADASYN, do not provide the same value as real data. A common issue with oversampling methods is their potential to fail in introducing genuinely informative instances, which can result in overfitting. This overfitting occurs when the classifier becomes too tailored to synthetic examples, reducing its generalization ability to unseen data [18].

**2.2.2 Undersampling.** Unlike oversampling, undersampling reduces the number of instances in the majority class to create a balanced dataset. One of the simplest methods is random undersampling, but a common issue with this approach is that it ignores the information contained in the instances that are removed. Ensemble techniques, such as EasyEnsemble and BalanceCascade [19], address this by iteratively undersampling the majority class while training multiple models. Yen et al. [20] proposed cluster-based undersampling approaches for selecting the representative data as training data. This ensures that the undersampling process still reflects the variability of the majority class. Sampling within each cluster can be done in various ways, by random selection or based on the average distance between instances in each cluster and the minority class samples [20–22]. Huang et al. [23] introduced a Hybrid Neural Network with Clustering-Based Undersampling Technique, which segments the majority class into clusters and selects representative nodes to balance the dataset without sacrificing essential information.

Lin et al. [24] proposed two strategies for clustering-based undersampling. The first is the Cluster Centers Strategy, which sets the number of clusters equal to the number of minority class instances. The centroids (or cluster centers) of these clusters represent the majority class, reducing its size to match that of the minority class. The second strategy, Nearest Neighbors of Cluster Centers, selects the nearest real data points to the cluster centers, rather than the centroids, to represent the majority class. This approach ensures the use of actual data points, as centroids may not always correspond to real instances.

Undersampling methods are typically divided into two categories: (i) fixed undersampling and (ii) cleaning undersampling [25]. These methods differ in how they handle the balance of the dataset, with some ensuring a completely balanced dataset and others leaving it partially imbalanced. Many traditional classifiers struggle when applied to imbalanced datasets, especially those that rely heavily on class distributions, use accuracy in the optimization process, or use it as an evaluation metric. In such cases, fixed undersampling methods are more appropriate. Examples of these methods include random undersampling, NearMiss [26], clustered centroids [24], Adaptive K-means clustering [27] and classification accuracy-based methods such as instance hardness threshold [28]. On the other hand, cleaning undersampling methods focus on deleting instances that meet certain conditions, which are not directly tied to achieving a specific balancing ratio but rather aim to clean the feature space based on empirical criteria. Examples of these approaches include condensed nearest neighbors [29], Edited Nearest Neighbor (ENN) [30], Nearest Neighborhood Cleaning rule (NNC) [31], one-sided selection [32], and Tomek Links [33].

The ENN operates by considering each sample in the majority class and its k nearest neighbors. If the majority class sample's label disagrees with the majority of its neighbors, it is removed from the dataset. This process reduces overlap between classes, making the decision boundaries clearer to improve prediction [30].

Undersampling and oversampling methods can also be combined. A hybrid algorithm combining ENN with oversampling methods, such as SMOTE, has been used to enhance prediction in medical datasets [34,35]. This approach balances datasets by first cleaning noisy instances with ENN, then synthetically generating samples from the minority class. ENN can also be used along with ADASYN, as has been demonstrated on lysine succinylation predictions [36]. The NNC, a variation of ENN, refines the removal process by taking three neighbors into account. [37] has applied this in software defect prediction and other domains.

Tomek Links, introduced by Tomek [33], works by identifying borderline instances that are close to another class, helping to refine decision boundaries. A pair of samples $(x_i, x_j)$ from different classes form a Tomek Link if $d(x_i, x_j) < d(x_i, x_k)$ for all other $x_k$ in the dataset, with $d$ typically denoting Euclidean distance. Instances forming a Tomek Link are considered

ambiguous, leading to the removal of the majority class instance in the link. Consequently, class overlap is reduced and class boundaries are sharpened.

We believe that undersampling should be carefully managed to remove samples that are easily distinguishable from the target class while retaining those that are similar or closer to it, which are samples that are harder to identify. Much like humans preparing for an exam, training with harder questions increases the likelihood of solving easier ones. Many successful algorithms emphasize focusing the learning process on "harder instances" from earlier iterations. For example, boosting methods like Adaboost assign greater weight to misclassified instances, increasing the likelihood that these instances will be prioritized for classification in subsequent iterations. Over time, this approach allows the model to focus on samples that are more difficult to classify, as they provide the most valuable feedback for refining decision boundaries.

A related concept is highlighted in [38], where optimal sampling of the majority class is achieved by assigning more weight to samples with a higher probability of being positive. These high-probability samples can be identified using an artificial neural network model trained on randomly undersampled data, referred to as a "Copilot model". The process involves filtering these higher-probability samples to enhance the performance of another model. This adaptive behavior is also the foundation of ADASYN oversampling, which generates synthetic samples for minority-class instances that are harder to classify, focusing on samples near the class boundary.

The effect of uncertainty sampling on a simple logistic regression model is further illustrated in [39]. This implies that retaining samples that confuse the model is beneficial as it forces the model to focus on distinguishing true positive samples from negative samples that resemble positives. We explored approaches to address imbalance, aiming to improve training for detecting minority samples and introduced an effective solution, which will be introduced next.

## 3 Research methodology

This section begins with a description of the dataset and preparatory techniques, followed by a brief review of supervised and unsupervised algorithms. It concludes with the resampling implementation and the parameters used in the study.

### 3.1 Data preprocessing

The petroleum dataset contains five continuous-type measurements from logging tools, which record physical and mechanical properties of different rock types and enclosed fluids. Resistivity Log (LLD) measures the electrical resistance of the rock. One end of the tool sends an electric current into the formation and the other records the voltage difference [1]. Rocks naturally emit gamma radiation due to the presence of radioactive elements, Gamma Ray Log (GR) detects and quantifies this radiation. Neutron Log (NPHI) works by emitting high-energy neutrons into the surrounding formation, when these neutrons collide with hydrogen nuclei (found in water or hydrocarbons), they lose energy and are kept away from the tool's receiver, which measures the number of slowed-down neutrons. Density Log (RHOB) tool measures the density of the formation: gamma rays are emitted from a radioactive source, and their interactions with the electrons in the rock are used to estimate density, by calculating the amount of scattered radiation. Sonic Log (DTC) measures the travel time of acoustic waves through the rock. The target feature represents hydrocarbon bearing zones obtained by the geological method. Each log alone cannot differentiate between different fluid or rock types. By combining data from multiple logging tools, geologists can accurately interpret subsurface [1,40].

Non-normally distributed features were log-transformed prior to standardization. Test data was scaled using the scaling parameters (mean and standard deviation) derived from the training data. All models use the same training set, test set, random seeds and hyperparameters to ensure a fair comparison and highlight the added value. Only 3.5% of samples in the training dataset (three wells) are labeled as positives. The test dataset, consisting of two independent wells, is used to validate predictions and detect overfitting. It shares the same characteristics as the training data.

## 3.2 Supervised classifiers

Supervised classifiers are ML algorithms designed to map input data to corresponding labeled outputs based on a provided dataset. SVC, Random forest, Adaboost, XGBoost and KNN are effective because they balance simplicity and complexity. Unlike linear or naive algorithms, they are sophisticated enough to avoid underfitting, yet they are less computationally demanding than deep learning, making them suitable for scenarios without massive datasets or extensive computing resources.

## 3.3 Unsupervised algorithms

Unsupervised learning algorithms are crucial for revealing hidden statistical structures, providing novel insights from complex non-linear relationships. KMeans was used to partition data into k clusters, k being set by the user as the number of centroids. The algorithm minimizes intra-cluster variance, iterating between assigning points to centroids and updating them.

Gaussian Mixture Models (GMM) assume that data is generated from a combination of multiple Gaussian distributions, each representing a different cluster. The model characterizes each Gaussian component by its mean and covariance matrix. These parameters are estimated using the Expectation Maximization (EM) algorithm, which iteratively computes the probability of each data point belonging to each cluster and then updates the parameters to maximize these probabilities [41].

Agglomerative clustering is a hierarchical, bottom-up clustering method that starts with each point as its own cluster and merges the closest clusters iteratively. The process uses a proximity matrix based on a similarity metric, commonly Euclidean distance, to identify the nearest pairs for merging. Key parameters include the choice of distance metric and linkage criterion, such as single, complete, or average linkage, which influence how cluster distances are calculated [42].

Another approach used is Density-Based Spatial Clustering of Applications with Noise (DBSCAN), which identifies clusters based on density, making it effective for detecting arbitrary-shaped clusters and handling noise. It grows clusters from core points that have at least Minimum number of Points (MinPts) neighbors within an epsilon radius ($\epsilon$), while sparse points are classified as noise [42]. Clustering algorithms were employed in the baseline setup, since conducting a comprehensive grid search for the optimal parameter combination was beyond the scope of the paper and might be impractical for resampling comparison applications.

## 3.4 Resampling variants

The imbalanced-learn library (version 0.13.0) in Python is specifically designed to address imbalanced datasets [25,43]. The `sample_strategy` parameter, available in all undersampling functions, specifies which majority class to resample; in this study, the default setting 'auto' was used. Techniques such as Tomek Links, random undersampling, and clustered centroids lack significant adjustable parameters that would notably impact the results. ENN and NNC demonstrate slightly higher instances removal from the majority class with larger `n_neighbors` values. Various parameters were investigated with respect to their corresponding performances at Sect 4.3.

# 4 Results

This section includes a description of Progressive Clustering Undersampling (PCU) and examines its impact on prediction performance compared to other resampling methods under varying levels of noise and class imbalance.

## 4.1 Data resampling

The PCU workflow presented in S3 Fig is inspired by the idea that clustering algorithms can guide us toward similar instances in a multidimensional feature space. By alternating multiple clustering stages with simple supervised learners, we progressively focus on the target class in specific 'saved' steps, rather than direct training with a supervised algorithm.

This approach will be particularly effective for detecting rare anomalies of interest. PCU's process is visualized on the petroleum dataset in Fig 1(a).

The training set undergoes a series of clustering steps (Fig 1(a)). At each stage, the target instances progressively converge into a single unique cluster. The cluster containing nearly all minority samples is retained, while the others are discarded. This progressive within-clustering process continues until the minority instances can no longer be grouped into a single cluster, reaching what is referred to as the "rupture" or "spread-out" stage. The rupture stage is marked by an unsupervised decision boundary that divides the minority group nearly in half, signaling the formation of the last cluster. At this point, the last cluster is ready for a supervised algorithm to further refine its focus, distinguishing positives from negatives that closely resemble positives.

Initially, KMeans clustering was used on the training dataset, set to create 5 groups (see Fig 1(a)). This resulted in all positive target instances (722 in total) being placed in cluster 4. The second column presents the outputs of three clustering trials applied to cluster 4, with the bottom Sankey diagram corresponding to the best-performing model. This strategy reflects the idea that PCU is not confined to one specific unsupervised algorithm, DBSCAN initially treated the rare positives as noise. Several attempts with different $\epsilon$ and MinPts values were necessary to find the optimal density that effectively captured the positives. KMeans clustered 96.6% of the target instances into cluster 1, with the remaining 24 positive instances being likely outliers, located naturally away from most positive instances. The third approach (GMM) grouped 98% of the target instances into a single cluster, leaving only 14 as outliers. This result was similar to the second trial, with minimal clustering differences. Consequently, we proceeded with the third trial as the basis for further analysis.

In the next step, samples falling into the violet and red labeled clusters were removed while samples of the blue cluster were retained. This resulted in a more balanced dataset containing the positives (Fig 1(a), bottom line). The guiding principle for any clustering process is to select the most effective unsupervised algorithm or optimal hyperparameters to capture the majority of positives, as positives are expected to occupy a specific location within the feature space.

A third clustering step is performed using KMeans, where nearly all target instances are grouped into the yellow cluster (Cluster 1). Any further clustering of this unique cluster resulted in splitting the positive instances roughly in half (fourth clustering in Fig 1(a)). This further clustering becomes ineffective and serves as a natural stopping point. The last cluster (yellow) derived from the third clustering step includes nearly all positives along with a few negatives. S1 Fig offers an alternative route leading to comparable results.

Returning to the main route (Fig 1(a)), training on the labels of the yellow cluster (Cluster 1) from the third clustering phase leads to the development of the Semi-Guided Model. This cluster, represented by yellow dots in Fig 2, is visualized during each step of rightward undersampling process. Initially, the entire training set appears indistinct, complicating visual interpretation and the ability of supervised algorithms to make distinctions. However, the dataset becomes more manageable for a supervised model to learn from. Saving a location of the yellow cluster by training a classifier on the cluster labels always yields very high accuracy.

The spatial relationship between the majority and minority classes is visualized in the second-row subplots (Fig 2), which illustrate the distribution of samples across the RHOB and NPHI scaled features. These plots show how the PCU method removes the most distant instances from the target in stages, retaining those that are most similar. The first two PCA components are also visualized in the third-row subplots, with each removal wave showing a different PCA projection.

In the presented approach, each clustering stage on the training set is followed by training a supervised model to predict that stage's unique cluster. These models ensure the same routing behavior when applied on test data and simplifies the refinement process and the overall classification task. As illustrated in Fig 1(b), Model 1 predicts the unique cluster (baby blue), while Model 2 and Model 3 predict unique clusters for later stages. Model 3-dash, trained on the yellow cluster dataset, differs by specifically identifying positive instances rather than clusters. The PCU ensures that samples, which are confusing for traditional models, are retained, allowing Model 3-dash to effectively distinguish positives from similar

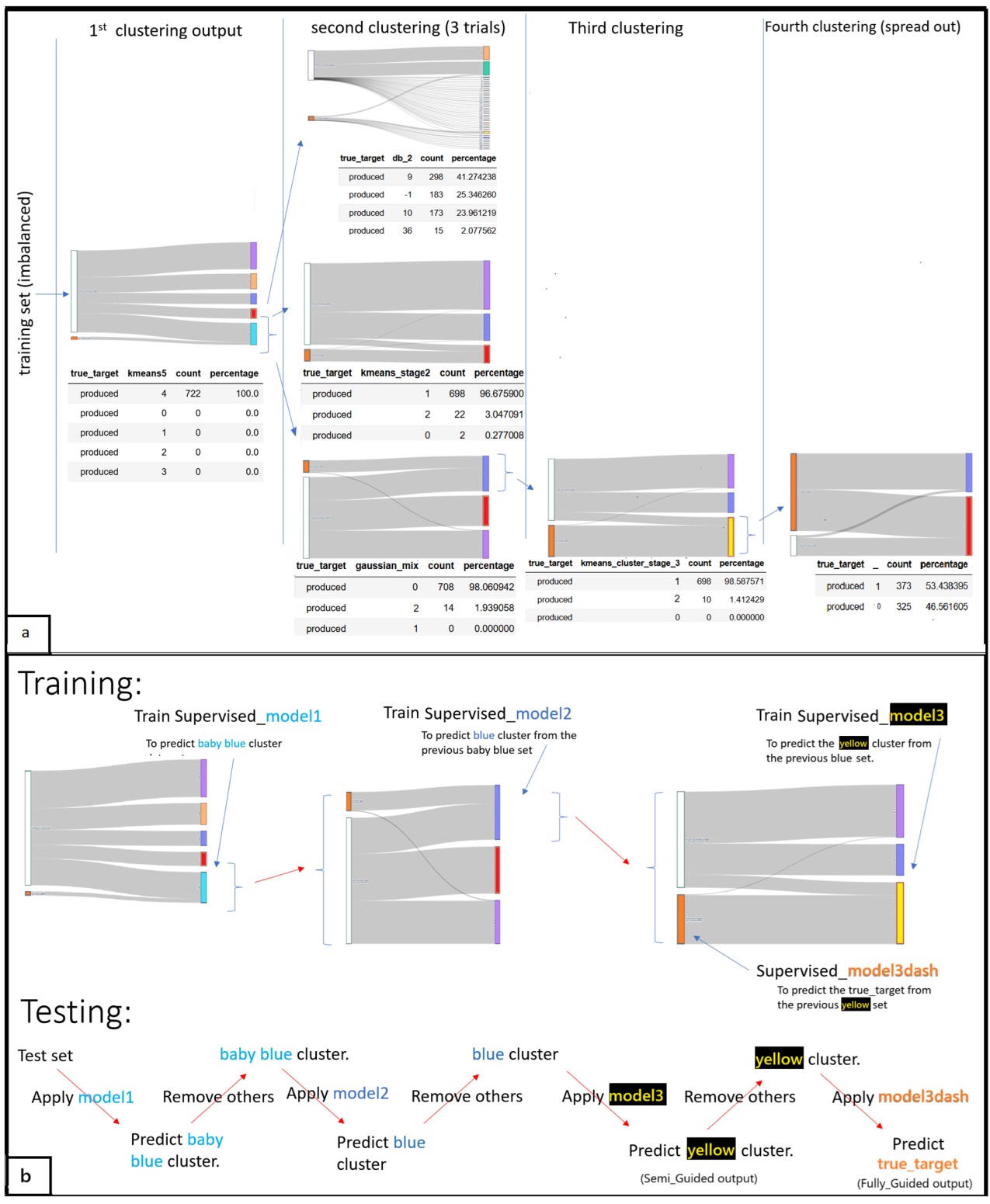

**Fig 1**. **(a) PCU applied to the petroleum dataset.** At each clustering stage, a single resulting cluster consistently contains the majority of positive samples, except at the final stage. (b) Supervised algorithms are used to guide the test set along the same path followed by the training set.

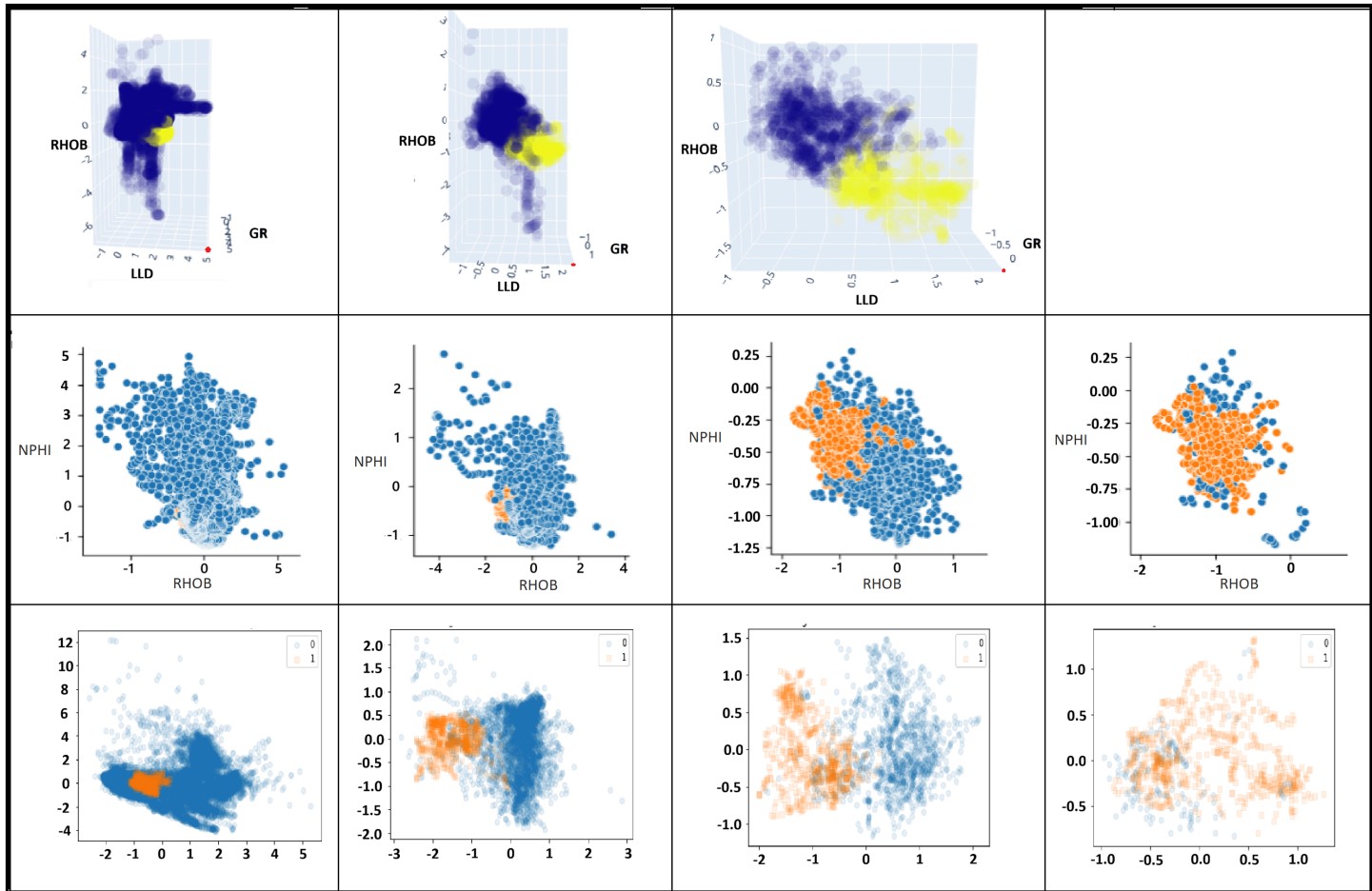

**Fig 2. The scatter plots illustrate the undersampling process.** The first column shows the original training set. The 3D plot uses scaled features, with the final cluster highlighted in yellow. The second row distinguishes minority (orange) and majority (blue) classes, and the third row uses the first two Principal Component Analysis (PCA) for visualization. Focusing on relevant instances by zooming in simplifies the subsequent training process.

negative samples (last column in Fig 2). When models are executed sequentially on the test dataset, they provide two important outputs:

1. Prediction from Semi-Guided Branch (Model 3 in Fig 1(b)).
2. Prediction from Fully-Guided Branch (Model 3-dash in Fig 1(b).

The Semi-Guided Branch involves training a supervised algorithm to identify the naturally defined group that contains nearly all positive instances. In this method, training is focused on the specific cluster containing these instances, rather than on explicit positive class labels. However, the resulting predictions on the test dataset are treated as predictions of positive instances and evaluated accordingly. The Fully-Guided Branch of PCU involves training a supervised model to detect positives within the final cluster, thereby creating a tighter boundary around them. As a result, the Fully-Guided output consistently achieves higher precision, while the Semi-Guided output offers greater recall.

To visually compare our method with other State-of-the-art (SOTA) resampling techniques, Fig 3 illustrates the first two PCA of the dataset after applying each technique. Fixed undersampling methods reduced the majority class by 19,913

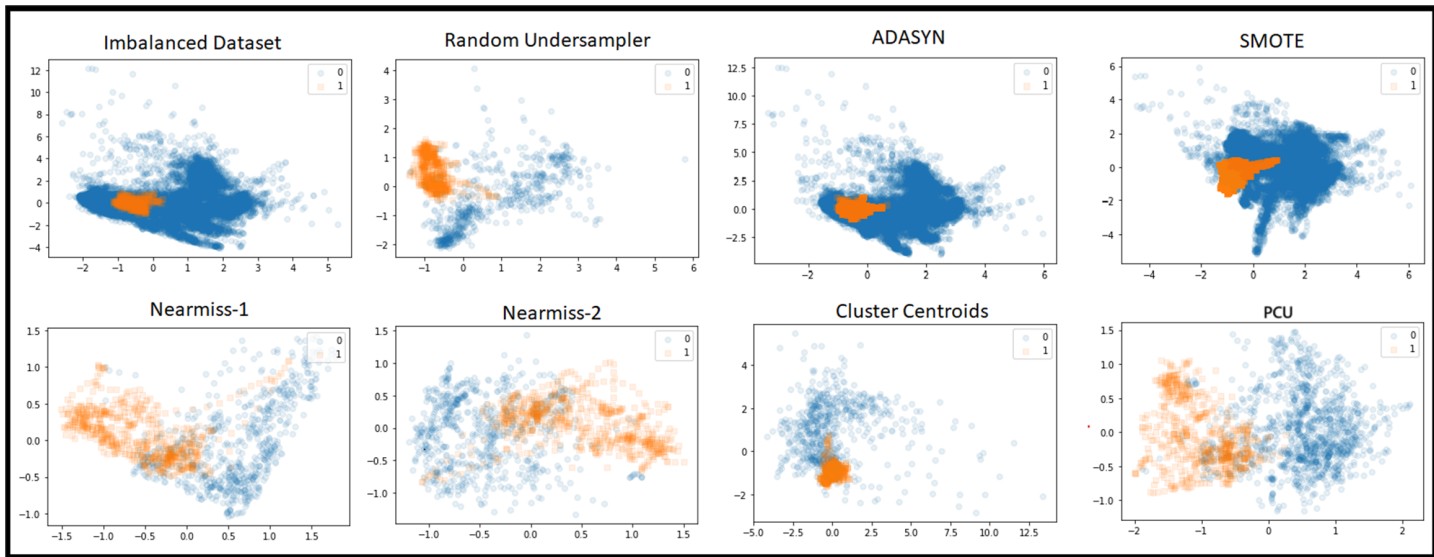

**Fig 3. The first two PCA components of the training data before and after applying balancing methods.** The best separation between classes occurs by PCU method after the second removal wave.

instances, leaving both classes with 722 instances each. In case of cleaning undersampling methods, which do not enforce balance, showed significantly less reduction. Tomek Links, ENN, and NNC removed only 6, 76, and 51 instances, respectively, so their PCA visualization resembles the original dataset. Oversampling methods such as SMOTE and ADASYN balanced the classes by expanding the minority class size from 702 to 20,655 instances. The balanced state of PCU reveals a clearer separation of target instances compared to other balancing methods. This enhanced separation, achieved through the unsupervised algorithm, leads to higher recall and precision.

## 4.2 Modeling

The occurrence of hydrocarbon bearing layers was predicted by KNN classifier in the test wells and compared with the target feature and logging responses (Fig 4). In the case of PCU, each unique cluster prediction from different stages is plotted in tracks displayed from right to left (first-clustering, second-clustering), culminating in the Semi-Guided and Fully-Guided outputs. These correspond to model1, model2, model3, and model3-dash respectively. Notice the progressive filtering stages, moving leftward toward the Fully-Guided track. At each stage, the instances most dissimilar to the target are filtered out, effectively implementing a controlled undersampling process (Fig 4). It is important to note that some resampling methods, like NearMiss1 and NearMiss2, produce many false positives, while others result in false negatives when compared to the geological detection method (Fig 4).

## 4.3 Evaluation

When different supervised algorithms are trained without resampling, their performance scores vary significantly (Fig 5). However, performance variations are more pronounced when comparing the resampling effect applied with different classifiers.

Regarding F1-scores, the Semi-Guided model outperformed others in both classifiers, achieving 65% with KNN and 60% with Random Forest. The non-resampled KNN attained 56%, while Random Forest recorded the lowest score at

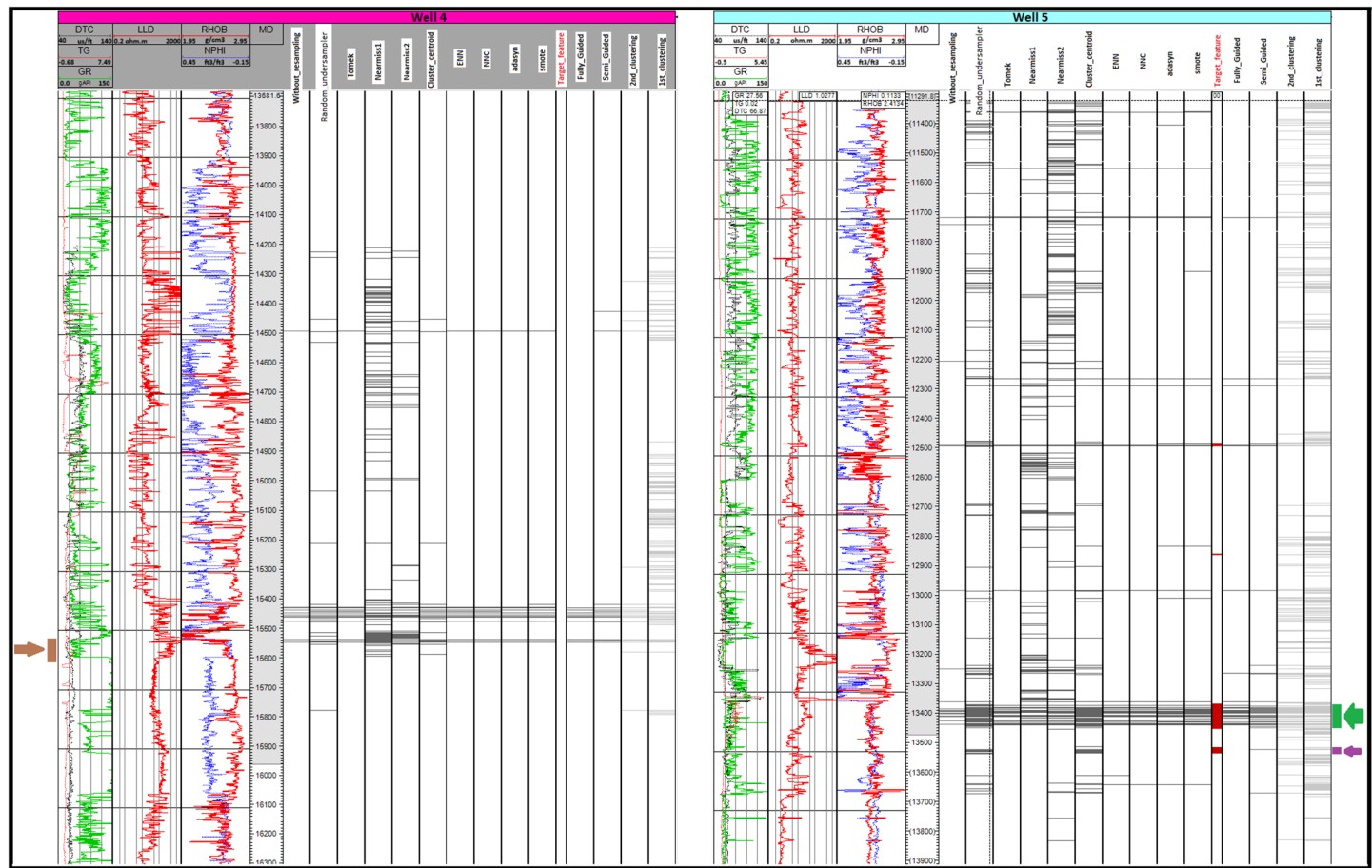

**Fig 4**. Logging responses for the two test dataset wells, along with prediction outputs (grey) and target layers (red), Measured Depth (MD) represents the depth within the well [feet].

10%. In terms of precision, the Fully-Guided model achieved the highest results with both KNN (75%) and Random Forest (73%). The non-resampled KNN yielded 70%, whereas Random Forest reached 39%. These results demonstrate the effectiveness of the new method and suggest that Random Forest particularly benefits from PCU-guided balancing.

As this study investigates the impact of resampling methods on classification performance, it is also important to examine key resampling parameters to evaluate potential performance variations. S1 Table presents alternative resampling parameters, deviations from the default settings used in the study, changes in sample counts, and their corresponding evaluation metrics. For instance, setting the number of neighbors to 100 removes a sample if most or all of its 100 nearest neighbors belong to the target class, resulting in a slight change in the F1-score. Overall, the parameter variations led to no substantial differences compared to the primary settings discussed earlier in the study.

## 4.4 Examining noise, balance and arbitrary shapes

The two moons dataset is a synthetic, two-dimensional dataset consisting of two interleaving half circles, serving as a classic benchmark for evaluating classification algorithms on non-linearly separable data. To assess the effectiveness of resampling methods under real-world conditions, noise was introduced before testing various methods. "Noise" in this context refers to minority instances incorrectly labeled as majority class. Fig 6 illustrates three versions of the training set,

| Condition | Algorithm | Recall | Precision | Fscore |
|---|---|---|---|---|
| Without resampling | Random Forest | 0.06 | 0.39 | 0.10 |
| | Adaboost | 0.16 | 0.44 | 0.24 |
| | KNN | 0.46 | 0.70 | 0.56 |
| | XGBoost | 0.09 | 0.40 | 0.14 |
| | SVC | 0.25 | 0.72 | 0.37 |
| Resampling with KNN | Semi-Guided | 0.60 | 0.70 | 0.65 |
| | Fully-Guided | 0.49 | 0.75 | 0.59 |
| | Random Undersampler | 0.89 | 0.40 | 0.55 |
| | Tomek | 0.48 | 0.70 | 0.57 |
| | Nearmiss1 | 0.51 | 0.14 | 0.22 |
| | Nearmiss2 | 0.46 | 0.16 | 0.24 |
| | Cluster Centroids | 0.87 | 0.48 | 0.62 |
| | ENN | 0.50 | 0.68 | 0.57 |
| | NNC | 0.49 | 0.68 | 0.57 |
| | ADASYN | 0.52 | 0.68 | 0.59 |
| | SMOTE | 0.54 | 0.67 | 0.60 |
| Resampling with Random Forest | Semi-Guided | 0.51 | 0.72 | 0.60 |
| | Fully-Guided | 0.32 | 0.73 | 0.45 |
| | Random Undersampler | 0.39 | 0.48 | 0.43 |
| | Tomek | 0.05 | 0.38 | 0.09 |
| | Nearmiss1 | 0.28 | 0.28 | 0.28 |
| | Nearmiss2 | 0.17 | 0.24 | 0.20 |
| | Cluster Centroids | 0.24 | 0.50 | 0.32 |
| | ENN | 0.07 | 0.39 | 0.12 |
| | NNC | 0.07 | 0.36 | 0.11 |
| | ADASYN | 0.06 | 0.37 | 0.10 |
| | SMOTE | 0.09 | 0.49 | 0.15 |

**Fig 5**. **Evaluation scores for various models on the test set include classifiers without resampling, as well as SOTA resampling methods combined with KNN and Random Forest classifiers.** PCU achieves the highest F1-score in the Semi-Guided setting and the highest precision in the Fully-Guided setting.

along with the common test set used for all evaluations. Version one exhibits the least class imbalance and minimal label noise, while these issues are most pronounced in Version three. All versions represent varying degrees of label uncertainty. Specifically, Versions one, two, and three contain 42, 18, and 6 minority class labels, versus 278, 302, and 314 majority class instances respectively.

PCU was applied to the noisy versions of the dataset. Table 1 summarizes the parameters of the unsupervised algorithms tested on each version, along with the resulting unique cluster capacity for positive instances and the degree of undersampling achieved. A single round of clustering proved sufficient, as additional attempts disrupted the minority class cluster. Among the methods tested, the DBSCAN algorithm—configured with parameters $\epsilon = 0.2$ and MinPts = 9—was the most effective for isolating target instances. It meets the two required conditions; produced a cluster containing 97.60% positive instances and enabled the largest removal of majority class samples (253).

After identifying the optimal clustering approach for PCU, we compared the performance of a KNN classifier on datasets resampled either by this optimal PCU method or by other resampling techniques (see Table 2). The Fully-Guided model maintains its optimal precision up to moderate noise levels (dataset Versions one and two) but drops to zero with high noise and severe class imbalance (Version three). In contrast, the Semi-Guided branch of PCU sustains high performance even under high-noise conditions (Versions two and three). In such scenarios, classifying based on boundaries defined by unsupervised clustering algorithms is more effective than training a classifier on unreliable labels or extremely

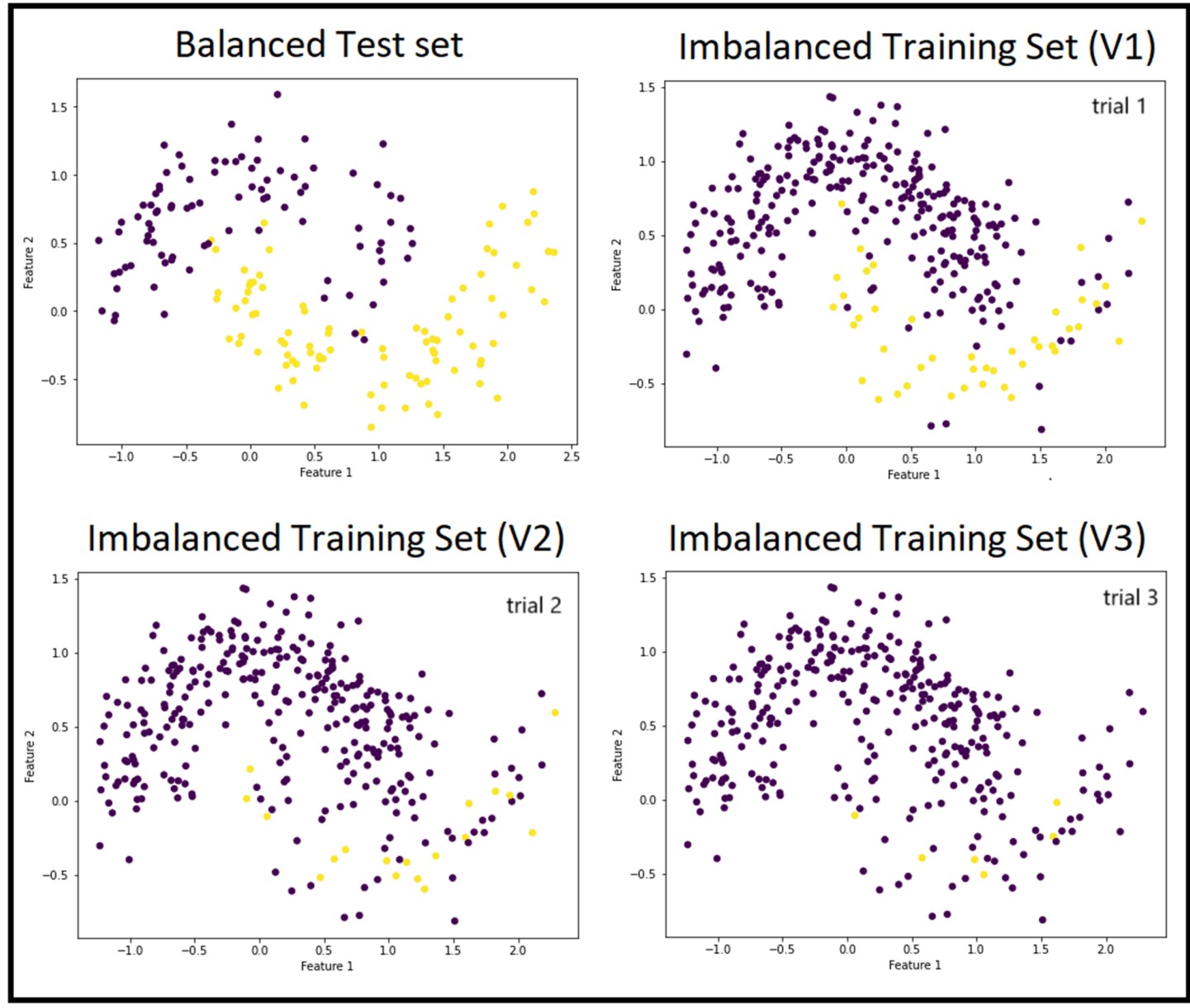

**Fig 6**. **Two moons dataset with three training sets of different noise levels and a test set for evaluation.**

rare positives. The method most similar to the Semi-Guided approach for the tested datasets is clustered centroids, as both rely on unsupervised algorithms. However, our approach offers the advantage of preserving samples most similar to the target class, whereas clustered centroids prioritize diversity, aiming to retain the overall distribution and structure of the original data.

The versions were tested using different train/test splits with 5 random seeds. The alternative splits produced nearly identical results. Pairwise paired t-tests were conducted between the Semi-Guided method and each of the other ten

**Table 1**. **Settings for unsupervised algorithms applied to various noisy versions of the Two Moons dataset, showing the percentage of minority class samples retained within the unique cluster and the number of removed samples.** The algorithm in bold indicates the selected approach for further evaluation.

| Clustering method | Minority samples kept [%] | | | Samples removed |
|---|---|---|---|---|
| | V1 | V2 | V3 | |
| KMeans (k = 2) | 78.57 | 83.00 | 83.00 | 164 |
| Agglomerative (k = 2) | 97.60 | 100.00 | 100.00 | 181 |
| Agglomerative (k = 3) | 97.61 | 100.00 | 100.00 | 181 |
| GMM (k = 2) | 100.00 | 100.00 | 100.00 | 124 |
| GMM (k = 3) | 97.60 | 100.00 | 100.00 | 220 |
| GMM (k = 4) | 71.00 | 83.00 | 83.00 | 272 |
| DBSCAN ($\varepsilon = 0.1$, MinPts = 10) | 100.00 | 100.00 | 100.00 | 13 |
| DBSCAN ($\varepsilon = 0.15$, MinPts = 10) | 97.60 | 100.00 | 100.00 | 177 |
| DBSCAN ($\varepsilon = 0.2$, MinPts = 10) | 97.60 | 100.00 | 100.00 | 240 |
| DBSCAN ($\varepsilon = 0.25$, MinPts = 8) | 57.00 | 50.00 | 50.00 | 33 |
| **DBSCAN ($\varepsilon = 0.2$, MinPts = 9)** | 97.60 | 100.00 | 100.00 | 253 |

**Table 2**. **KNN classifier average performance with standard deviation on versions of the Two Moons dataset resampled using either PCU or other techniques.** The Semi-Guided method significantly reduces the impact of noise on the subsequent KNN classifier. Symbol †: Statistically significant compared to Semi-Guided method. Symbol ‡: Statistically significant compared to Fully-Guided method.

| Resamp. Method | V1 F1 | V1 Prec. | V2 F1 | V2 Prec. | V3 F1 | V3 Prec. |
|---|---|---|---|---|---|---|
| Semi-Guided | $0.93 \pm 0.06$ | $0.98 \pm 0.02$ | $0.93 \pm 0.06$ | $0.98 \pm 0.02$ | $0.93 \pm 0.06$ | $0.98 \pm 0.02$ |
| Fully-Guided | $0.78 \pm 0.09$ | $0.99 \pm 0.01$ | $0.27 \pm 0.05$ | $1.00 \pm 0.00$ | $0.00 \pm 0.01$ | $0.20 \pm 0.45$ |
| none | $0.81 \pm 0.06$ | $0.99 \pm 0.01$ | $0.28 \pm 0.05†$ | $1.00 \pm 0.00$ | $0.01 \pm 0.02†$ | $0.40 \pm 0.55$ |
| Random Resamp. | $0.97 \pm 0.01$ | $0.98 \pm 0.02$ | $0.93 \pm 0.06$ | $0.95 \pm 0.04$ | $0.84 \pm 0.05$ | $0.87 \pm 0.06$ |
| Tomek Links | $0.90 \pm 0.05$ | $0.99 \pm 0.01$ | $0.40 \pm 0.06†$ | $1.00 \pm 0.00$ | $0.04 \pm 0.06†$ | $0.40 \pm 0.55$ |
| nearmiss(1) | $0.81 \pm 0.07$ | $0.95 \pm 0.08$ | $0.54 \pm 0.08†$ | $0.72 \pm 0.16$ | $0.41 \pm 0.08†$ | $0.39 \pm 0.04$ |
| nearmiss(2) | $0.75 \pm 0.01†$ | $0.61 \pm 0.02‡$ | $0.74 \pm 0.04$ | $0.63 \pm 0.07‡$ | $0.57 \pm 0.05†$ | $0.77 \pm 0.22$ |
| Clustered Centroids | $0.97 \pm 0.01$ | $0.97 \pm 0.01$ | $0.94 \pm 0.07$ | $0.96 \pm 0.02$ | $0.85 \pm 0.04$ | $0.90 \pm 0.04$ |
| ENN | $0.96 \pm 0.02$ | $0.99 \pm 0.01$ | $0.67 \pm 0.08†$ | $1.00 \pm 0.00$ | $0.09 \pm 0.09†$ | $0.60 \pm 0.55$ |
| NNC | $0.95 \pm 0.02$ | $0.99 \pm 0.01$ | $0.62 \pm 0.07†$ | $1.00 \pm 0.00$ | $0.09 \pm 0.09†$ | $0.60 \pm 0.55$ |
| ADASYN | $0.95 \pm 0.02$ | $0.97 \pm 0.01$ | $0.83 \pm 0.05$ | $0.98 \pm 0.02$ | $0.73 \pm 0.09†$ | $0.95 \pm 0.02$ |
| SMOTE | $0.94 \pm 0.02$ | $0.98 \pm 0.01$ | $0.83 \pm 0.05$ | $0.98 \pm 0.02$ | $0.72 \pm 0.08†$ | $0.96 \pm 0.02$ |

methods for F1-scores, and between the Fully-Guided method and the remaining methods for precision. A Bonferroni correction was applied to adjust for multiple comparisons. Both Semi-Guided and Fully-Guided demonstrated statistically significant improvements over certain methods at the 0.05 significance level, even after correction (Table 2).

Similar experiment was conducted where the noise in the training features was increased to the point that the two crescents were no longer visually distinguishable (V4, V5 and V6 in Fig 7). Note that this type of noise was introduced before splitting. The Fully-guided approach offered no additional benefit compared to simple training without resampling. However, Semi-guided PCU and clustered centroids performed the best among the resampling methods.

## 5 Discussion

Our experiments demonstrate the effectiveness of PCU by evaluating it on both synthetic and real-world datasets and observing its stable and repeatable performance. These results suggest that PCU is inherently domain-agnostic and well-suited for scientific and sensor-based environments, where physical measurements frequently produce continuous, high-dimensional data streams. Such settings naturally benefit from methods capable of uncovering subtle, unlabeled anomalies embedded within complex feature patterns.

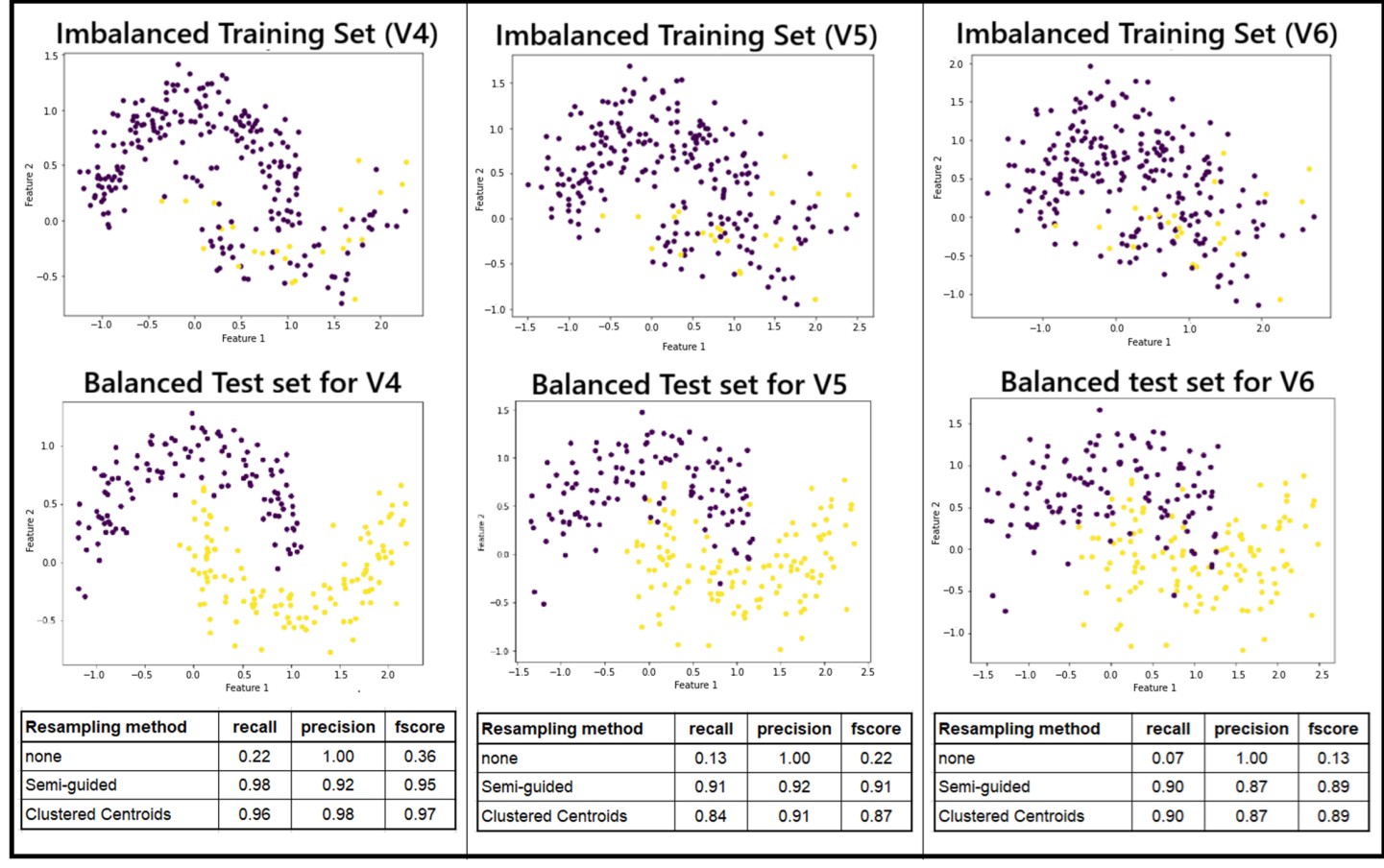

**Fig 7**. **Two moons dataset with three training sets of different feature noise levels and test sets for evaluation.**

The value of the described PCU method lies in using unsupervised ML to define decision boundaries based on the spatial distribution of clusters. This supports supervised learners by preserving the location of relevant clusters after each clustering step. 'Preserving' means that supervised learners can more easily learn from clustered groups, resulting in higher accuracy. A key aspect of this approach is choosing the appropriate unsupervised algorithm with optimal parameters to define the best boundaries. Achieving this may require several iterations as described in Table 1.

### 5.1 Limitations and challenges

Minority class samples often occupy specific regions of the feature space, forming distinct patterns such as clouds, spheres, or multidimensional Gaussian distributions. In this context, similar rock types are expected to produce comparable logging responses. When logging tools frequently pass through hydrocarbon-bearing layers, the corresponding responses naturally cluster within the dataset.

The effectiveness of PCU relies heavily on the number of samples available per class. In the training dataset, transitions between hydrocarbon-bearing reservoirs (positives) and water-bearing reservoirs (negatives)—which are the only two options—occur less frequently than fully saturated fluid reservoirs. These rare transitions allow unsupervised algorithms to identify and form boundaries between these states. Consequently, if the dataset only includes transitions

between different fluid types, the models may struggle to delineate them effectively. Thus, it is crucial to ensure the training data has a sufficient number of target instances to form clear patterns for optimal performance with the proposed method. This forms the core assumption: even when labels are sparse or unreliable, an underlying structure or pattern still exists within the feature space.

Another limitation is that PCU is built on the assumption—that the dataset is imbalanced and contains noise. During undersampling, some target instances may be removed despite their rarity; such outlier positives would otherwise be expected to mislead a model in a traditional setting without resampling. Experiments on a benchmark dataset (the Iris dataset, originally introduced by [44]), where 33% of the training data belongs to the Iris Versicolor species and is designated as the minority target class, show that neither PCU nor any resampling technique outperforms using the original training set without resampling. This is because the dataset is inherently simple—free of outliers, low in noise, and only mildly imbalanced (S2 Fig).

## 5.2 Encapsulating PCU

The experimentation required to best capture the target pattern can be carried out either through a customized version of the PCU workflow or a simple ready-to-use implementation. Each choice has its advantages and limitations. The widespread nature of the problem may make automation preferable to minimize coding as much as possible, encapsulating the entire process in pipelines would allow for automated selection of the best path through a pre-defined list of clustering algorithms. Up to this point, we have described the experimental, user-guided setup—executed step by step—where the user tests multiple unsupervised algorithms or parameter settings, selects the best cluster inclusion from sankey diagrams, and ultimately performs undersampling.

### Algorithm 1 PCU

```
 1: Select the list of clustering algorithms along with their parameters.
 2: Define the maximum inclusion threshold for the rupture stage (X).
 3: repeat
 4:   Execute the clustering algorithms on the training set.
 5:   Compute the inclusion of positive samples within each cluster. If any cluster's inclusion
      exceeds (X), continue; otherwise, exit the loop.
 6:   For each cluster, compute its inclusion multiplied by the number of instances that would be
      removed if it were selected. Identify the unique cluster with the highest resulting value.
 7:   Train a supervised model to learn the location of this cluster in the feature space.
 8:   Replace the training set with the selected cluster.
 9: until No cluster has an inclusion value exceeding X (i.e., rupture).
10: Train a supervised model on the final cluster using the true positive labels (model-dash).
11: Apply the sequence of saved models to the test data.
12: Evaluate the performance of the models on the test data.
```

A fully automated PCU algorithm is summarized in the pseudo-code shown in Algorithm 1. This only requires two user-defined inputs: a list of unsupervised algorithms and the maximum inclusion threshold for the rupture stage. Default settings using GMM (with $k = 3, 4$, and $5$) generally serves the purpose well. Setting the second parameter to 70% is also typically effective, as the drop value in cluster inclusion that defines the rupture stage usually does not exceed this value. PCU is not tied to a specific unsupervised algorithm because it bases its decisions solely on the inclusion values of the resulting clusters, irrespective of the algorithm that produced them.

Each cluster in every clustering stage would be tested against two variables: the amount of cluster inclusion and the amount of removed instances. These can be combined in a function, with (optionally) controlled weights for each factor. The simplest function involves multiplying them to ultimately select the cluster with the higher product value, then proceeding to the next undersampling stage, examining same listed clustering algorithms and so on (Algorithm 1).

Utilizing DBSCAN introduces a more tailored version of the workflow. For example, if the domain knowledge expects that the target cluster has an arbitrary shape, it is advisable to experiment with DBSCAN using different parameter combinations to accurately capture the cluster's density, rather than setting a predefined list of clustering algorithms. This involves selecting appropriate MinPts and $\epsilon$ values. Since feature spaces in real-world scenarios often contain clusters with varying densities, parameter tuning becomes essential. This challenge can be addressed by advanced applications of DBSCAN, as suggested by [45–47], which offer solutions for managing varying cluster densities.

### 5.3 Dealing with uncertainty

The ground truth approach to identifying hydrocarbon reservoirs involves perforation (Well 5) or core sampling (Well 4). A 15-foot section at a depth of 13,530 feet in Well 5 (Fig 4, purple arrow) revealed a low hydrocarbon presence, insufficient for production. Although this constitutes a geological false positive, both the Fully-Guided and Semi-Guided models correctly predicted it as negative. Another segment in the same well was found to be productive (Fig 4, green arrow).

The interval between 15,540 and 15,600 feet in Well 4 was investigated using conventional core sampling and confirmed to be non-productive, as predicted by most models (brown arrow in Fig 4). The primary advantage of this approach is the ability to select the best-performing model or calibrate it to ground truth data, effectively functioning as a multidimensional ruler or standardized tool—thereby reducing the need for costly additional investigations on future wells.

PCU integrates both supervised and unsupervised differentiation. This relationship is illustrated in S3 Fig. The green and violet boundaries were derived from unsupervised algorithms, drawn through gaps between instances. The green boundary, which encloses most positive instances along with some negatives, is particularly valuable when leveraged by a supervised model to produce a Semi-Guided output. If we have higher confidence in the positive labels within the training set, training on the target labels confined by the green boundary can improve precision—this represents the Fully-Guided approach

In real-world scenarios where some positive instances may be uncertain, prioritizing the identification of all positives can be more important, depending on the relative costs of false positives versus false negatives [48] or domain-specific requirements. The Semi-Guided approach is especially valuable when a secondary screening process, involving more advanced tools, follows the initial detection of positive candidates. It is also highly effective in dealing with imbalanced datasets—for example, in a case with a 2% minority class such as the Two Moons (Version three) dataset, the Semi-Guided method improves precision, recall, and F1-score by approximately 10% (Table 2). In such contexts, the Semi-Guided approach is a safer and more reliable choice as the primary predictive model.

Geologists can more effectively infer the statistical characteristics of hydrocarbon zones from the final cluster in the training data than by relying solely on positive instances. This improves the differentiation between positive cases and similar negative ones, enabling more accurate probability calculations for measurement distributions.

## 6 Conclusion

Detecting minority classes in imbalanced datasets is challenging, as learning algorithms often prioritize majority class patterns, resulting in bias due to insufficient minority representation. This study introduces a novel method that simplifies datasets by reducing both outliers and instances farthest from the minority class, thereby achieving a more balanced dataset. It demonstrates that minority class samples can be detected without solely relying on the shape and distribution of the positive class. Instead, event frequency helps define decision boundaries, identified through unsupervised methods. The workflow employs clustering to retain similar instances in multidimensional spaces, progressively refining toward the target class. By alternating clustering with supervised learning, representative of the minority class are effectively captured, as evidenced by higher performance metrics of subsequently applied classifiers.

Three evaluation metrics were assessed on two datasets, each evaluated with and without resampling using common supervised algorithms. In addition, we tested PCU with four unsupervised approaches and two routing paths through

undersampling stages, exploring different balance and noise levels using state-of-the-art resampling methods with varying parameters and random runs. These variations in criteria and conditions were applied to ensure a comprehensive and fair comparative performance analysis.

When using the KNN classifier with various resampling methods in the petroleum dataset, PCU achieved the highest F-score at 65%, followed by the Clustered Centroids undersampling method at 62%. For Precision, PCU again led with 75%, while the next best was Tomek Links at 70%. Using a Random Forest classifier, PCU obtained an F1-score of 60%, outperforming Random Undersampling at 43%. In terms of Precision, PCU reached 73%, followed by Clustered Centroids at 50%. PCU demonstrates clearer separation in the PCA components for the real-world dataset compared to other approaches. Its consistent performance on the Two Moons dataset indicates that PCU is domain-agnostic and well-suited for scientific and sensor-based data, where physical measurements frequently produce high-dimensional data streams.

The study recommends using the Semi-Guided approach in scenarios with high noise, uncertain labeling, or severe class imbalance. This method leverages natural grouping, occurrence frequency, and distance relationships rather than directly training on rare positive instances, which are prone to overfitting. It is particularly well-suited for situations where missing positive cases is costly—such as missing hydrocarbon-bearing zones after drilling—by emphasizing recall.

The Fully-Guided approach is recommended when the cost of false positives is high, like adjusting well operations for an assumed positive zone that turns out to be false. Together, these components ensure robust detection of the minority class samples. Both outputs should be reported to define the risk range, allowing decision-makers to perform effective risk assessments.

The study equips ML practitioners with the methodology to extract clarity and value from noisy, imbalanced and high-dimensional data, enhancing prediction and leading to more effective resource allocation through investments or cost savings.

## Supporting information

**S1 Fig. Another possible path for PCU process.** The final result doesn't change much.
(PNG)

**S1 Table. KNN classification performance after using training datasets resampled with under- and oversampling methods.** Each resampling method was tested with different resampling parameters. Provided are the respective resampling parameters (other than the standard settings), resampling outputs and associated evaluation scores. As an example, Nearmiss-1 was tested with three alternative parameters which reduced the majority class from 20,635 to 722. Training on these resampled instances produced the corresponding evaluation scores on the test dataset.
(xlsx)

**S2 Fig. Evaluation scores for different models measured from Iris test set.** Grey models are supervised classifiers trained directly without resampling. Orange models represent KNN executed after different resampling methods. KNN without resampling is sufficient to achieve the desired results for a simple dataset.
(PNG)

**S3 Fig. Study flow and predictive models generated using various combinations of resampling and classifier techniques.** SOTA resampling methods, along with the newly introduced PCU, were evaluated on multiple datasets. The PCU workflow integrates clustering alternated with classification steps to produce Semi-Guided and Fully-Guided decision boundaries.
(PNG)

## Acknowledgments

The authors gratefully acknowledge Khalda Petroleum Company and the Egyptian General Petroleum Corporation for providing the essential data and materials for this study. We also express our sincere appreciation to IU University of Applied Sciences for offering the academic environment and resources that made this research possible.

## Author contributions

**Conceptualization:** Amr Abuzeid.

**Data curation:** Amr Abuzeid.

**Formal analysis:** Amr Abuzeid.

**Methodology:** Amr Abuzeid.

**Resources:** Amr Abuzeid.

**Supervision:** Elena Jolkver.

**Validation:** Amr Abuzeid.

**Visualization:** Amr Abuzeid.

**Writing – original draft:** Amr Abuzeid.

**Writing – review & editing:** Amr Abuzeid, Elena Jolkver.

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
