## [Decision Letter · Decision Letter 0]

27 Oct 2025

PONE-D-25-38473Rare Event Detection by Progressive Clustering UndersamplingPLOS ONE

Dear Dr. Abuzeid,

Thank you for submitting your manuscript to PLOS ONE. After careful consideration, we feel that it has merit but does not fully meet PLOS ONE’s publication criteria as it currently stands. Therefore, we invite you to submit a revised version of the manuscript that addresses the points raised during the review process.

The reviewers have completed the reviewing, and you could find the comments by the reviewers. It is required to enhance its scientific rigor, clarity, and completeness before it can be considered for acceptance. The key revisions needed are:

Methodological and Experimental Rigor: It is necessary to elaborate on the specific implementation details of the PCU algorithm, including the criteria for selecting clustering algorithms, determining key parameters (e.g., the "rupture" point), and conducting parameter sensitivity and computational complexity analyses. Please also supplement the description of the experimental setup and incorporate statistical significance tests to strengthen the credibility of the comparative results.

Depth of Analysis and Discussion: The manuscript requires a dedicated subsection to systematically discuss the limitations of the proposed method, potential failure scenarios, computational costs, and its generalizability to other domains (e.g., finance, healthcare). It is also recommended to include a summary table clearly outlining the advantages and disadvantages of related studies.

Presentation and Language Quality: Please enhance the resolution and clarity of the figures (Figures 1-4) and ensure the "Graphical Abstract" mentioned is either included or all references to it are removed. Furthermore, a thorough revision of the text is needed to address language issues, reference formatting (which requires updating), and consistency in terminology (e.g., the "Semi-Guided" branch).

I look forward to receiving your revised manuscript accompanied by a detailed point-by-point response letter. I hope the reviewers' feedback is useful to you.

We look forward to receiving your revised manuscript.

Kind regards,

Hui Li

Academic Editor

PLOS ONE

Journal Requirements:

2. Please note that PLOS One has specific guidelines on code sharing for submissions in which author-generated code underpins the findings in the manuscript. In these cases, we expect all author-generated code to be made available without restrictions upon publication of the work. Please review our guidelines at https://journals.plos.org/plosone/s/materials-and-software-sharing#loc-sharing-code and ensure that your code is shared in a way that follows best practice and facilitates reproducibility and reuse

Reviewers' comments:

Reviewer's Responses to Questions

**Comments to the Author**

1. Is the manuscript technically sound, and do the data support the conclusions?

Reviewer #1: Yes

Reviewer #2: Yes

2. Has the statistical analysis been performed appropriately and rigorously?

Reviewer #1: Yes

Reviewer #2: Yes

3. Have the authors made all data underlying the findings in their manuscript fully available?

Reviewer #1: Yes

Reviewer #2: Yes

4. Is the manuscript presented in an intelligible fashion and written in standard English?

Reviewer #1: Yes

Reviewer #2: Yes

5. Review Comments to the Author

Reviewer #1: this study explores various resampling techniques and introduces a novel method called Progressive Clustering Undersampling (PCU). This technique removes negative instances that are distant from positive ones. PCU was compared with eight common undersampling and two oversampling techniques, consistently outperforming them on highly imbalanced and noisy datasets. The workflow demonstrates that rare anomalies can be effectively predicted using unsupervised methods based on frequency-driven decision boundaries. Progressive clustering ultimately identifies clusters with the highest concentration of positive instances. These delineated clusters are then saved by supervised models and used in the preparatory phase before prediction. The proposed method produces two outputs: one optimized for a high F1- score and the other for high precision. Overall, this approach presents a promising solution for identifying rare anomalies in complex, imbalanced data environments.

Good work keeps up

But some comments are needed?

all of them are submitted to the editor

Reviewer #2: Overall Evaluation

This manuscript presents a novel approach named Progressive Clustering Undersampling (PCU) for detecting rare events in highly imbalanced datasets. The study is well-motivated and addresses an important challenge in machine learning and data-driven anomaly detection. The integration of clustering-based unsupervised learning with progressive undersampling and supervised refinement is innovative and potentially impactful.

The manuscript is generally well-written, methodologically sound, and supported by comprehensive experiments on both real-world and synthetic datasets. However, several key areas require major revisions to improve clarity, reproducibility, and scientific rigor before the paper can be accepted for publication.

Major Comments

Methodological Clarity and Reproducibility

The PCU algorithm is interesting but described in a largely conceptual way. The pseudocode in Algorithm 1 lacks specific implementation details such as:

How clustering algorithms are selected or switched between stages.

Criteria used for determining the “rupture” point.

Parameter tuning strategies for clustering algorithms.

Please expand this section with more formal definitions, computational complexity, and parameter sensitivity analysis.

Comparative Evaluation

While the paper includes comparisons with several resampling methods, it would strengthen the work to include statistical significance tests (e.g., t-test or Wilcoxon) across repeated runs.

Please clarify whether the same random seed or data splits were used across all methods to ensure fairness.

Generalizability

The petroleum dataset is well described, but the method’s generalizability beyond this specific application (e.g., finance, healthcare, cybersecurity) should be better discussed.

Consider adding one more benchmark dataset from a different domain to show robustness.

Figures and Visualizations

Figures 1–4 are valuable but require higher resolution and clearer legends.

The “Graphical Abstract” mentioned in the text is missing from the submission. Please include it or remove all references to it.

Ablation Study

Since PCU combines unsupervised clustering and supervised refinement, it is important to show the contribution of each step individually (e.g., clustering-only, supervised-only, and combined).

Language and Structure

Some sections (e.g., Introduction and Related Work) are overly descriptive. Please focus more on critical comparisons and concise synthesis.

Check for consistency in reference formatting (e.g., DOI style and numbering).

Minor Comments

Typographical and formatting errors occur in several places (e.g., “undersam-pling,” “pos-itive”). Please revise carefully.

Include clear variable definitions when first introduced in equations or pseudocode.

The term “Semi-Guided” and “Fully-Guided” branches should be formally defined and consistently referenced.

Ensure all URLs in the references are correctly formatted and accessible.

Clarify whether “noise” in the Two Moons dataset refers to label noise or feature noise.

6. PLOS authors have the option to publish the peer review history of their article (what does this mean?). If published, this will include your full peer review and any attached files.

Reviewer #1: No

Reviewer #2: No

---

## [Author Response · Author response to Decision Letter 1]

10 Dec 2025

Dear Editor,

I would like to sincerely thank you and the reviewers for your valuable time, insightful comments, and constructive suggestions and modifications. Your feedback has been instrumental in improving the clarity and quality of this manuscript. We have carefully addressed all points raised, and believe these revisions have greatly enhanced the overall writing.

All additions or changes in the marked-up copy appear in red, while deletions are shown with strikethrough. Answers here include searchable keywords, written in italics.

Reviewer One:

1. Need to add subsection for more explanation about the limitation, challenges, contribution and structure of the article.

Performed in subsec. 5.1.

structure of the article :

‘The rest of the paper is structured ..’

2. It is better to add table summarizing the x-studies showing the advantages and disadvantages.

I definitely agree with you that summarizing the related studies in a table would be better. However, we chose to keep the comparison within the text rather than summarizing it because of limitations on the number of tables and pages. We reserved those for the methodology description and results.

3. A detailed analysis of the limitations and potential failure scenarios of the proposed model is missing.

Detailed in subsec. 5.1.

4. Some information about datasets and experiment setup is needed.

Subsec. 3.1 (one page) describes the petroleum dataset. while the second dataset was described in:

‘The two moons dataset is a synthetic, ….’

please refer whether a specific information is missing so we can modify or enhance writing.

experiment setup description is enhanced in subsec. 5.2.

5. Additional comparative analysis, around computational requirements, cost and robustness of the model with other SOTA methods.

‘Three evaluation metrics were assessed on two datasets’…

Regarding the computational aspect, here is our perspective:

PCU simply integrates one or more common supervised and unsupervised algorithms to undersample the training set. The computational complexity is tied to the specific algorithms used, with their exact complexity functions typically outside the scope of this work and usually obtainable from the foundational algorithm papers.

In general, undersampling methods may require time to reduce the number of samples, but they offer clear advantages during model building compared with oversampling techniques.

6. Provide quantitative remarks of the impact of the proposed method in the abstract and conclusion.

The quantitative results were found to be too extensive to include in the abstract due to character limitations. Listing the scores of PCU branches compared to the next best resampling methods for both datasets, in terms of F-score and precision, would exceed the allowable length. Therefore, we instead summarized these findings in the conclusion:

‘When using the KNN classifier with various resampling …’

7. References needed to be updated (24/25).

Updated with related articles.

Reviewer Two

Methodological Clarity and Reproducibility

The PCU algorithm is interesting but described in a largely conceptual way. The pseudocode in Algorithm 1 lacks specific implementation details such as:

• How clustering algorithms are selected or switched between stages.

A list of user-defined algorithms required to run the PCU and can be used in all stages for simple implementation. PCU compares between clusters and select the unique one regardless of its origin:

‘PCU is not tied to a specific unsupervised algorithm …’,

Criteria used for determining the “rupture” point.

This was answered in:

‘until the minority instances can …’ to ‘formation of the last cluster’.

in addition, clearly stated in the added subsec. 5.2

We Also:

- Explained farther the pseudo-code (Algorithm 1).

- Differentiated between customized version of PCU and automated one.

• Parameter tuning strategies for clustering algorithms.

We focused more on testing parameters of resampling algorithms, since the study’s main goal is to compare different resampling methods. Tuning unsupervised parameters would be extensive and could lead to a huge number of trials when combined to other options. This is illustrated in:

‘Clustering algorithms were employed in the baseline setup …’.

Please expand this section with more formal definitions, computational complexity, and parameter sensitivity analysis.

All were described in the new subsec. 5.2.:

Regarding the computational complexity. Here is what we think:

PCU simply integrates one or more common supervised and unsupervised algorithms to undersample the training set. The computational complexity is tied to the specific algorithms used, with their exact complexity functions typically outside the scope of this work and usually obtainable from the foundational algorithm papers.

In general, undersampling methods may require time to reduce the number of samples, but they offer clear advantages during model building compared with oversampling techniques.

Comparative Evaluation

While the paper includes comparisons with several resampling methods, it would strengthen the work to include statistical significance tests (e.g., t-test or Wilcoxon) across repeated runs.

Five repeated runs with different splits were reported to ensure that the results are not occurred by chance. Table 2 was modified to include various runs results.

‘Noise versions were tested using different train/test splits ….’

Please clarify whether the same random seed or data splits were used across all methods to ensure fairness.

This is illustrated in:

‘All models use the same training set, test set ….’

Generalizability

The petroleum dataset is well described, but the method’s generalizability beyond this specific application (e.g., finance, healthcare, cybersecurity) should be better discussed. Consider adding one more benchmark dataset from a different domain to show robustness.

Incorporating an additional industry dataset would be very useful but also would greatly expand the paper beyond our page limit. We have outlined our expectations regarding other domains where PCU may be beneficial within the paragraph:

‘Our experiments demonstrate the effectiveness of …’

Additionally, A benchmark dataset was experimented and added to enhance limitation understanding:

‘Another limitation is that PCU is built on the assumption—that the dataset’

Figures and Visualizations

Figures 1–4 are valuable but require higher resolution and clearer legends.

Modified all, Thank you.

The “Graphical Abstract” mentioned in the text is missing from the submission. Please include it or remove all references to it.

We apologize for the inaccessibility. It was in the original draft at the bottom of the manuscript; we moved its placement after the abstract, since the figure is typically uploaded separately in editorial managers.

Ablation Study

Since PCU combines unsupervised clustering and supervised refinement, it is important to show the contribution of each step individually (e.g., clustering-only, supervised-only, and combined).

The contribution of the unsupervised component is illustrated using both Sankey diagrams and cluster inclusion tables. And

‘displayed from right to left ….’

In figure 4. Also displayed as

‘green and violet boundaries …’

In the Graphical abstract.

The performance of the supervised component is reported in the petroleum dataset evaluation, without resampling (Figure 5) and in the Non-resampled results (Table 2). The combined model (i.e., PCU) is evaluated under the ‘Fully-guided’ and ‘Semi-guided’ in both the corresponding figure and table.

Language and Structure

Some sections (e.g., Introduction and Related Work) are overly descriptive. Please focus more on critical comparisons and concise synthesis.

We summarized some, and removed others (marked by strikethrough).

Check for consistency in reference formatting (e.g., DOI style and numbering).

Modified, Thank you.

Minor Comments

Typographical and formatting errors occur in several places (e.g., “undersam-pling,” “pos-itive”). Please revise carefully.

Hyphenation settings at lines ends can be modified according to the journal settings.

Include clear variable definitions when first introduced in equations or pseudocode.

Clearer definitions were added specifically in the Algorithm definition (pseudocode). Its placement was moved ahead to Subsec. 5.2, to leverage the definitions from Section 4—such as cluster inclusion, the number of removals, and the rupture stopping point.

The term “Semi-Guided” and “Fully-Guided” branches should be formally defined and consistently referenced.

‘The Semi-Guided Branch involves training ..’

Ensure all URLs in the references are correctly formatted and accessible.

Done, thank you.

Clarify whether “noise” in the Two Moons dataset refers to label noise or feature noise.

The Two Moons dataset includes both types. Additionally, we added experiments that vary each type individually. Table 2 presents more test variations in label noise, while Figure 7 was added to evaluate variations in feature noise.

‘Similar experiment was conducted where the noise’

We appreciate your time and consideration, and we hope that the revised version meets the expectations of the reviewers and the editorial board

---

## [Decision Letter · Decision Letter 1]

28 Dec 2025

Rare Event Detection by Progressive Clustering Undersampling

PONE-D-25-38473R1

Dear Dr. Abuzeid,

We’re pleased to inform you that your manuscript has been judged scientifically suitable for publication and will be formally accepted for publication once it meets all outstanding technical requirements.

Kind regards,

Hui Li

Academic Editor

PLOS One

Additional Editor Comments (optional):

Reviewers' comments:

Reviewer's Responses to Questions

**Comments to the Author**

1. If the authors have adequately addressed your comments raised in a previous round of review and you feel that this manuscript is now acceptable for publication, you may indicate that here to bypass the “Comments to the Author” section, enter your conflict of interest statement in the “Confidential to Editor” section, and submit your "Accept" recommendation.

Reviewer #1: All comments have been addressed

Reviewer #2: All comments have been addressed

2. Is the manuscript technically sound, and do the data support the conclusions?

Reviewer #1: Yes

Reviewer #2: Yes

3. Has the statistical analysis been performed appropriately and rigorously?

Reviewer #1: Yes

Reviewer #2: Yes

4. Have the authors made all data underlying the findings in their manuscript fully available?

Reviewer #1: Yes

Reviewer #2: Yes

5. Is the manuscript presented in an intelligible fashion and written in standard English?

Reviewer #1: Yes

Reviewer #2: Yes

6. Review Comments to the Author

Reviewer #1: This study presents Capturing rare events in severely imbalanced datasets is challenging, as the learning and optimization processes are often biased toward the majority class. To address this issue, this study explores various resampling techniques and introduces a novel method called Progressive Clustering Undersampling (PCU). This technique removes negative instances that are distant from positive ones. PCU was compared with eight common undersampling and two oversampling techniques, consistently outperforming them on highly imbalanced and noisy datasets. The workflow demonstrates that rare anomalies can be effectively predicted using unsupervised methods based on frequency-driven decision boundaries. Progressive clustering ultimately identifies clusters with the highest concentration of positive instances. These delineated clusters are then saved by supervised models and used in the preparatory phase before prediction. The proposed method produces two outputs: one optimized for a high F1- score and the other for high precision. Overall, this approach presents a promising solution for identifying rare anomalies in complex, imbalanced data environments.

Good work keeps up

Reviewer #2: Recommendation: Accept with Minor Revisions

Overall Assessment

The authors have done an excellent job addressing the concerns raised in the previous round of reviews. The manuscript has been substantially improved in terms of clarity, methodological detail, reproducibility, and discussion of limitations. The introduction of subsection 5.1 (limitations/challenges/contributions/structure) and 5.2 (experimental setup, complexity, parameter choices) directly resolves many of the earlier criticisms. Figures have been upgraded, the graphical abstract is now properly placed, and additional experiments on noise variations strengthen the claims. The responses to both reviewers are thoughtful, transparent, and courteous.

The proposed Progressive Clustering Undersampling (PCU) method is novel, well-motivated for highly imbalanced and noisy settings, and the empirical evidence (across petroleum and Two Moons datasets) convincingly shows superior performance over a wide range of established resampling baselines.

The paper is now suitable for publication in PLOS ONE after only minor revisions.

Minor Revisions Required

Abstract – Quantitative Summary

The abstract states that PCU “consistently outperform[s]” other methods but provides no quantitative indication. While the authors correctly note character limits, consider adding a brief quantitative remark, e.g.:

“achieving up to X% higher F1-score and Y% higher precision than the next-best resampling technique on highly imbalanced noisy datasets.”

Even an approximate range or “substantial improvements” would help readers quickly gauge impact.

Conclusion – Strengthen Quantitative Takeaway

The conclusion mentions KNN results but could more prominently highlight the best overall gains of PCU (e.g., from Tables/Figures) across classifiers and datasets. A single sentence summarizing the magnitude of improvement would reinforce the contribution.

Statistical Significance

The authors added multiple runs and report variability, which is appreciated. For the final version, please consider adding a brief statement (or footnote to tables) indicating whether differences were statistically significant (e.g., via paired t-test or Wilcoxon on the repeated runs). This is not mandatory for PLOS ONE but would further strengthen the comparative claims.

Generalizability Discussion

The added benchmark discussion and limitation paragraph are helpful. To further improve readability, consider moving or duplicating the key sentence about expected beneficial domains (finance, healthcare, cybersecurity) into the Conclusion for emphasis.

Minor Editorial / Consistency Issues

Check for any remaining automatic hyphenations (e.g., “undersam-pling”, “pos-itive”) and disable if possible or manually correct.

Ensure all new references added in this revision follow the exact PLOS ONE style (especially DOI formatting).

In the text, “rupture point” is now clearer, but consider using a more standard term (e.g., “stopping criterion” or “termination condition”) or defining it explicitly on first use for broader accessibility.

Graphical Abstract

Confirm that the graphical abstract is uploaded as a separate high-resolution file in the final submission system, as required by PLOS ONE.

7. PLOS authors have the option to publish the peer review history of their article (what does this mean?). If published, this will include your full peer review and any attached files.

Reviewer #1: No

Reviewer #2: No

---

## [Editor Report · Acceptance letter]

PONE-D-25-38473R1

PLOS One

Dear Dr. Abuzeid,

I'm pleased to inform you that your manuscript has been deemed suitable for publication in PLOS One. Congratulations! Your manuscript is now being handed over to our production team.

Kind regards,

on behalf of

Professor Hui Li

Academic Editor

PLOS One